# Distinct Role of γ-Synuclein in the Regulation of Motor Performance and Behavioral Responses in Mice

**DOI:** 10.3390/biomedicines14010092

**Published:** 2026-01-02

**Authors:** Iuliia S. Sukhanova, Kirill D. Chaprov, Olga A. Morozova, Ruslan K. Ovchinnikov, Olga A. Kukharskaya, Valeria N. Zalevskaya, Nadezhda M. Yusupova, Anastasia A. Lugovskaya, Natalia N. Ninkina, Michail S. Kukharsky

**Affiliations:** 1Institute of Physiologically Active Compounds at Federal Research Center of Problems of Chemical Physics and Medicinal Chemistry, Russian Academy of Sciences, 142432 Chernogolovka, Russia; sukhanova.js@gmail.com (I.S.S.); chaprov@ipac.ac.ru (K.D.C.); morozovaoa@ipac.ac.ru (O.A.M.); rusovc@mail.ru (R.K.O.); loa.ipac@yandex.ru (O.A.K.); valeriya.nikolaevna2000@yandex.ru (V.N.Z.); 2Department of General and Cell Biology, Faculty of Medical Biology, Pirogov Russian National Research Medical University, 117997 Moscow, Russia; nadyusu@yandex.ru (N.M.Y.); anastasialgvsk@mail.ru (A.A.L.); 3School of Biosciences, Cardiff University, Cardiff CF10 3AX, UK; ninkinan@cf.ac.uk; 4Department of Pharmacology and Clinical Pharmacology, Belgorod State National Research University, 308015 Belgorod, Russia

**Keywords:** γ-synuclein, knockout (KO) mice, motor performance, behavioral phenotyping, aging

## Abstract

**Background:** The three Synuclein family members (α-, β-, and γ-synuclein) are presynaptic proteins that regulate synaptic vesicle trafficking and thereby influence neurotransmitter release. Synucleins belong to a class of intrinsically disordered proteins and are prone to aggregation into pathological deposits, which may impair their physiological synaptic functions. Knockout (KO) mouse lines, commonly used to model synuclein depletion in the nervous system, reveal a range of phenotypes with different motor and behavioral deficits. However, given the high sequence homology and functional interplay among the three synucleins, the specific contribution of each family member to these phenotypes remains poorly understood. **Objective:** In this study, we conducted a comparative phenotypic analysis of γ-synuclein KO, α- and β-synuclein KO, and αβγ-synuclein KO mice. **Methods:** Mice were subjected to a battery of behavioral tests assessing motor activity and coordination, anxiety-like behavior, and spatial learning and memory. Synaptic vesicle proteins were analyzed in brain tissues using Western blotting. **Results:** We observed that knocking out γ-synuclein but not α- and β-synucleins reduces mouse lifespan and leads to sustained reduction in muscle strength implicating that γ-synuclein is essential for longevity and motor system function. Another consequence of γ-synuclein deficiency is altered anxiety-like behavior manifested as a diminished aversive response, while exploratory behavior and memory remain intact. The triple KO mice mirror γ-synuclein KO mice in some behavioral changes, including shortened lifespan, reduced muscle strength, and decreased anxiety-like behavior. However, the triple KO mice additionally exhibit hyperactivity, which is not present in the other groups. No changes in synaptic vesicle marker levels were detected, indicating that the observed motor and behavioral abnormalities are not attributable to impaired synaptic connectivity. **Conclusions:** Taken together, these findings demonstrate nonredundant functions of individual synuclein family members and highlight a distinct role of γ-synuclein in regulating motor performance and behavioral responses.

## 1. Introduction

Synucleins are highly conserved cytosolic proteins that play an important role in regulating synaptic transmission in the central nervous system [1,2,3]. The synuclein family comprises three structurally similar proteins, α-, β-, and γ-synucleins, predominantly expressed in neural cells, and the patterns of their localization in different brain regions partially overlap [4,5,6]. Synucleins share structural similarities, but the functional specialization of individual synucleins remains poorly understood.

α-Synuclein is the most extensively studied member of the family because it is involved in the pathogenesis of several neurodegenerative diseases. It is the major component of Lewy bodies in Parkinson’s disease and dementia with Lewy bodies. α-Synuclein is also found in senile plaques in Alzheimer’s disease [7,8,9]. Similarly, γ-synuclein has the capacity to aggregate forming pathological deposits identified in amyotrophic lateral sclerosis and dementia with Lewy bodies [10,11,12,13]. Moreover, γ-synuclein is implicated in oncogenesis, because its expression is increased in various types of tumors [14,15,16,17]. In contrast, β-synuclein is generally considered non-amyloidogenic due to the absence of the NAC domain in its sequence [18,19,20]. Furthermore, β-synuclein can inhibit α-synuclein aggregation [21,22,23].

Transgenic mice lacking one, two, or all three synuclein family proteins are widely used to study biological functions of synucleins [24,25,26,27]. Some studies report that deletion of one or several synuclein genes does not result in significant impairments: animals remain fertile, exhibit no expressed morphological or behavioral deviations, and their lifespan is comparable to the lifespan of wild-type (WT) mice [28]. However, other studies report shortened lifespan and presence of a neurological phenotype with sensory dysfunction, such as age-related retinal degeneration and pronounced neurological deficits in triple KO mice [25,29].

The absence of overt phenotypic changes following synuclein inactivation suggests an auxiliary role of these proteins in nervous system physiology, with deficits becoming apparent only under specific conditions, such as during aging [29]. Indeed, aged synuclein-deficient mice exhibit statistically significant, although physiologically subthreshold alterations in the nigrostriatal system, including reduced dopamine levels [24]. Given that aging is accompanied by progressive functional decline in the central nervous system, the loss of one or more synucleins may exert a cumulative effect that becomes evident only at advanced age [25,29]. Supporting this hypothesis, more pronounced impairments have been reported in αβ-synuclein double KO mice compared to α-synuclein KO mice. Also, αβγ-synuclein triple KO mice demonstrated a more severe phenotype than αβ-synuclein double KOs [24,28].

Although structural similarity of synucleins implies potential functional redundancy, existing data indicate at least partial specialization within the synuclein family. β-Synuclein appears to play a primary role in regulating motor function: its absence leads to substantial coordination impairments, while additional loss of α- or γ-synuclein does not further worsen this phenotype [28]. Deletion of the α-synuclein gene is more strongly associated with behavioral alterations, including increased activation thresholds, reduced impulsivity [30], and impaired learning and memory [26,31]. Notably, α-synuclein KO mice do not differ from WT controls in anxiety-related behaviors [32]. In contrast, γ-synuclein KOs display reduced anxiety, enhanced exploratory activity, and accelerated adaptation to novel environments, suggesting that γ-synuclein may exert an inhibitory influence on cognitive processes [33]. These animals also exhibit enhanced working memory compared to WT mice, while long-term spatial memory performance in the Morris water maze remains unchanged [33]. Taken together, these findings indicate that α-synuclein KOs and γ-synuclein KOs exhibit oppositely directed alterations in memory and certain cognitive functions.

To elucidate the specific role of γ-synuclein in the development of motor and behavioral changes observed in synuclein-deficient mice, we performed a comparative analysis of motor and cognitive functions in γ-synuclein KO mice, α- and β-synuclein double KO mice, and αβγ-synuclein triple KO mice at 6, 13, and 24 months of age. Additionally, we quantified levels of some synaptic protein markers in brain tissues from these animals. The results may help clarify the role of γ-synuclein in neurodegenerative and psychiatric disorders characterized by pathological protein aggregation and consequent depletion of the functional protein pool in the cell, such as Parkinson’s disease and dementia with Lewy bodies. Furthermore, these results may inform therapeutic approaches aimed at preserving or compensating γ-synuclein function [16,34,35].

## 2. Materials and Methods

### 2.1. Animals

This study used mouse lines with constitutive knockout of synuclein genes: JAXMice 028559 B6(Cg)-Snca^tm1.2Vlb/J^ (SncaΔflox, Snca Δflox KO) with *Snca* deletion [36], JAXMice 006390 B6.129-Sncb^tm1.1Sud/J^ with *Sncb* deletion [24], and JAXMice 008843 B6.129P2-Sncg^tm1Vlb/J^ with *Sncg* deletion [37]. All KO lines were maintained on a C57BL/6J background. α- and β-synuclein double KO mice, and αβγ-synuclein triple KO mice were generated by intercrossing the corresponding single KO lines. C57BL/6J mice served as a WT control group. Animal colonies were maintained under standard vivarium conditions at a temperature of 22 °C on a 12/12 h light/dark cycle. Each experimental group included at least 20 animals. Exact sample sizes are indicated in the figure legends. Only male animals were used in all experiments to avoid the potential impact of female hormonal fluctuations on the results. All procedures adhered to the “Guidelines for accommodation and care of animals. Species-specific provisions for laboratory rodents and rabbits” (GOST 33216-2014) and complied with Directive 2010/63/EU on the protection of animals used for scientific purposes. The procedures were also approved by the local Institute Ethics Review Committee of the IPAC RAS (protocol No. 48, 15 January 2021). All reasonable measures were taken to limit the number of participating animals and minimize their suffering. All mice were genotyped by PCR using DNA obtained from ear biopsies, as described previously [27].

### 2.2. Behavioral Testing

Animals from all four experimental groups (γ-synuclein KO, αβ-synuclein KO, αβγ-synuclein KO, and WT) underwent a standardized battery of motor tests at 6, 13, and 24 months of age, including the grip-strength test, inverted-grid hanging test, and accelerating rotarod. At 13 months, exploratory behavior and psychoemotional state were additionally assessed using the open field, Y-maze, elevated plus maze, and Morris water maze tests. Animals were placed in the testing room 30 min before each experiment for habituation. Motor and cognitive tests were conducted under lighting conditions comparable to standard housing (~20 lx). The open field, Y-maze, and elevated plus maze tests were video-recorded and analyzed using EthoVision XT 11.5 software (Noldus, Wageningen, The Netherlands). The Morris water maze test was analyzed using ANY-maze 5.1 software (ANY-maze, Dublin, Ireland).

#### 2.2.1. Grip Strength Test

Mice were held by the tail and allowed to grasp a metal grid (7 × 12.1 cm, 6 × 6 mm spacing) attached to a grip strength meter ( IITC Life Sciences, Inc., Woodland Hills, CA, USA). The mean force across six trials was used for analysis [38].

#### 2.2.2. Inverted Grid Test

Mice were placed on an inverted metal grid (33.6 × 34.3 cm, 1.3 × 1.3 cm spacing), held 50–60 cm above a padded surface. The latency to fall was recorded, with a maximum test duration of 60 s. Mice that fell earlier received two additional trials at 15 min intervals. The best result of the three trials was used for analysis [39].

#### 2.2.3. Accelerating Rotarod Test

Mice were trained one day before testing at a constant speed (4 rpm) for 10 min. On the test day, mice were placed on a rod accelerating from 4 to 40 rpm over 4 min followed by 1 min at maximum speed (Ugo Basile 7650, Ugo Basile SRL, Gemonio VA, Italy). Each mouse completed four trials with 45 min intervals. The latency to fall was recorded [40].

#### 2.2.4. Open Field Test

Mice explored a 40 × 40 cm arena with 40 cm high walls (OpenScience, Moscow, Russia) under diffuse illumination (15 lx center, 10 lx corners). Activity was recorded for 5 min (Basler acA1300-60 camera, Basler AG, Ahrensburg, Germany) and analyzed using EthoVision XT 11.5 software (Noldus, Wageningen, The Netherlands). The following parameters were evaluated: total distance traveled, time spent in the center zone (20 × 20 cm), and number of center entries [41,42,43,44].

#### 2.2.5. Y-Maze

The maze consisted of three arms (40 × 6 × 20 cm) arranged at 120° angles (OpenScience, Moscow, Russia). The test included two 5 min sessions separated by 30 min. In session 1, two arms were open; in session 2, all three were open. Preference for the novel arm was calculated as the percentage of entries into the new arm during the first 2 min of session 2. Total distance, and time spent in each arm were also recorded [45].

#### 2.2.6. Elevated Plus Maze

The apparatus (OpenScience, Moscow, Russia) consisted of two open and two closed arms arranged at 90° around a central platform. Mice were placed in the center facing an open arm and allowed to explore for 5 min. Illumination at the center was 380–400 lx. Time and number of entries into open arms and distance and speed were recorded [46].

#### 2.2.7. Morris Water Maze

A circular pool, 150 cm in diameter, (OpenScience, Moscow, Russia) was filled with water to a height 60 cm above the base; water temperature was 22 °C. A hidden platform (10 cm diameter) was submerged 1 cm below the surface. Mice were trained over 4 days (4 trials/day; max trial length 60 s). On day 5, a 90 s probe trial was performed without the platform. The time to locate the platform and the number of platform-zone crossings were analyzed. Swim speed and total distance served as measures of physical activity [47].

### 2.3. Survival Analysis

After behavioral testing, animals were maintained under standard vivarium conditions until natural death. Lifespan (in days) was used to generate Kaplan–Meier survival curves and perform survival analysis.

### 2.4. Analysis of Synaptic Markers

Dorsal striatum and cerebral cortex samples were homogenized in 2× Laemmli buffer and heat-denatured at 95 °C for 5 min. Proteins were separated on 14% SDS-PAGE gels and transferred to PVDF membranes (Amersham, Sheffield, UK) by semi-dry transfer. Membranes were blocked with 4% non-fat dry milk in TBST (Tris-buffered saline with 0.1% Tween 20) and incubated with primary antibodies against synaptic proteins: synaptophysin (1:5000, S31620), synapsin IIa (1:10,000, 610666), and amphiphysin (1:10,000, A59420; all BD Biosciences; Transduction Laboratories, San Jose, CA, USA). HRP-conjugated secondary antibodies (1:4000, cat. 5178–2504, Bio-Rad, Hercules, CA, USA) were used for detection by enhanced chemiluminescence kit (ECL Plus, Thermo Fisher Scientific, Waltham, MA, USA) and X-ray film (Thermo Scientific, Waltham, MA, USA). Membranes were subsequently reprobed with α-tubulin antibodies (1:10,000, Proteintech, Rosemont, IL, USA) as a loading control.

### 2.5. Statistical Analysis

Statistical analysis was performed using GraphPad Prism 6 (GraphPad Software, San Diego, CA, USA). Data are presented as mean ± SEM unless otherwise indicated. The numbers inside the bars represent the number of animals per group. Normality was assessed using the D’Agostino-Pearson test. For normally distributed data, one-way ANOVA was applied; otherwise, the Kruskal–Wallis test was used. The statistical test and *p*-value threshold for each analysis are specified in the figure legends. Data analysis was performed in a blinded manner: the experimenters were unaware of group assignments until completion of the analysis.

## 3. Results

### 3.1. γ-Synuclein KO Mice Exhibit Reduced Lifespan

To assess the contribution of γ-synuclein to organismal viability, we conducted a comparative survival analysis of *Sncg* knockout (γ-KO) mice relative to WT controls. Two additional groups were included for comparison: mice with a double knockout of *Snca* and *Sncb* (αβ-KO), which retain an intact *Sncg* gene, and mice lacking all three synuclein genes (αβγ-KO). Analysis of the Kaplan–Meier curves demonstrated a statistically significant reduction in lifespan in γ-KO mice and in synuclein null mice (αβγ-KO), whereas αβ-KOs did not differ from WT controls (Figure 1).

Median survival (age at which 50% of animals remained alive) was 763 days in the γ-KO group, 859 days in the αβγ-KO group, 910 days in the αβ-KO group, and 913 days in the WT group. These findings indicate that loss of γ-synuclein and the combined loss of all synucleins negatively affect longevity, underscoring the importance of synucleins in maintaining organismal viability.

### 3.2. γ-Synuclein KO Mice Exhibit Decreased Muscle Strength

Motor functions and selected behavioral responses were evaluated in all KO groups compared to WT using the grip strength test, inverted grid test, and accelerating rotarod. At 6 months of age, which corresponds to young adult mice, γ-synuclein KO and αβγ-synuclein KO mice showed reduced forelimb grip strength compared with WT controls (Figure 2A).

In contrast, αβ-synuclein KO mice, which express γ-synuclein as the only remaining synuclein, displayed normal grip strength. Overall, mice lacking γ-synuclein (γ-synuclein KO and αβγ-synuclein KO) exhibited an approximately 30% reduction in strength relative to WT animals. Thus, γ-synuclein plays a key role in maintaining muscle strength, whereas α- and β-synucleins do not appear to compensate for its loss. In the inverted grid test, no statistically significant differences were observed among groups, although γ-synuclein KOs showed a trend toward shorter hang times (Figure 2B). These results suggest that short-term muscle endurance under dynamic load distribution is largely preserved in KO young adult mice. In the accelerating rotarod test, αβ-synuclein KOs exhibited a 28% reduction in fall latency compared with WT (Figure 2C) indicating reduced motor coordination and/or endurance. However, γ-synuclein KOs performed similarly to WT controls, which suggests that α- and β-synuclein, but not γ-synuclein, contribute more prominently to rotarod performance. Interestingly, αβγ-synuclein null mice showed increased fall latency relative to WT, possibly reflecting compensatory mechanisms resulting from the complete loss of all synucleins or increased activity in these animals (see below).

At 13 months (middle-aged mice), γ-synuclein KO and αβγ-synuclein KO mice again demonstrated reduced grip strength compared to WT mice, which is consistent with the 6-month findings (Figure 3A). αβ-synuclein KO mice were comparable to WT in grip strength. Also at 13 months, γ-synuclein KO mice exhibited a reduction in hang time in the inverted grid test, whereas αβ-synuclein KOs and αβγ-synuclein KOs did not differ from WT controls (Figure 3B). This finding indicates progressive impairment of motor endurance in γ-synuclein KO mice with age. In the accelerating rotarod test, only αβ-synuclein KO mice showed a trend toward reduced fall latency, and in contrast to the 6-month time point, αβγ-synuclein KO mice did not show increased performance (Figure 3C).

By 24 months (old age in mice), γ-synuclein KO mice no longer differed from WT animals in grip strength, although αβγ-synuclein KO mice still showed significantly reduced performance (Figure 4A). Notably, WT animals demonstrated age-related declines in grip strength (*p* = 0.0040, mixed-effects analysis with Tukey’s correction for 13 vs. 24 months), consistent with normal aging. These findings suggest that γ-synuclein loss reduces muscle strength at younger ages but does not exacerbate age-related decline, whereas complete synuclein deficiency has more severe long-term effects. In the inverted grid test, all KO groups demonstrated progressively reduced performance compared with WT (Figure 4B). In the accelerating rotarod test, the triple KO mice again showed increased fall latency, similar to the 6-month results, whereas γ-synuclein KO and αβ-synuclein KO mice did not differ from WT (Figure 4C). As in the grip strength test, WT mice demonstrated lower performance at 24 months (*p* = 0.0044, mixed-effects analysis with Tukey’s correction for 13 vs. 24 months).

### 3.3. γ-Synuclein KO Mice Display Reduced Anxiety-like Behavior and No Memory Impairments

To determine which behavioral responses beyond motor activity might be influenced by the loss of γ-synuclein we performed a series of tests assessing the general state of the nervous system and some higher neurological functions in middle-aged mice (13 months old). We first conducted the open field test. Overall, locomotor and exploratory activity were not altered in γ-synuclein KO mice. Parameters such as the total distance traveled and number of center entries also did not differ from control values (Figure 5A,C). In contrast, both the αβ-synuclein KO and triple KO mice exhibited significantly higher values for these parameters compared to WT. Interestingly, the hyperactivity observed in the triple KO mice is consistent with their motor test results. In contrast, the double KO mice manifested hyperactivity only in the open field test. It is possible that the exploratory component of activity is more pronounced than pure locomotion in αβ-synuclein KO mice. This is indirectly supported by a trend toward increased time spent in the center of the arena observed in the αβ-synuclein KO group compared to other groups (Figure 5B).

We performed the Y-maze test to assess short-term memory. No group differences were found for novel-arm exploration, and all animals showed a high preference (>40%) for the previously unexplored arm (Figure 6A) indicating intact short-term memory across groups. A hyperactive phenotype was again evident in the triple and double KO mice, reflected by an increased total distance traveled during the test (Figure 6B), which was consistent with the open field results.

Long-term memory was examined using the Morris water maze. All groups demonstrated a progressive decrease in latency to find the platform during training, which is indicative of normal learning (Figure 7A). Nonetheless, KO mice appeared to learn more slowly during the early days of training. Notably, γ-synuclein KO and αβγ-synuclein KO mice showed significantly lower swimming speeds (Figure 7C), which may indicate reduced muscle strength and endurance or diminished aversive response to water.

During the probe trial on day 5, all groups located the former platform area with similar speed and made a comparable number of crossings (Figure 7B,D). Thus, synucleins are likely not required for short- or long-term memory formation. Performance differences in the Morris water maze are more plausibly attributed to alterations in motor function or anxiety levels.

To assess anxiety-like behavior more specifically, we performed the elevated plus maze (EPM) test. γ-Synuclein KO and triple KO mice made more entries and spent significantly more time in the open and more stress-inducing arm of the maze compared to WT controls (Figure 8A,B), which indicates reduced anxiety-like behavior. As in the previous tests, triple KO mice exhibited a hyperactive phenotype—the total distance traveled was markedly greater compared to other groups (Figure 8C).

### 3.4. Absence of Synucleins Does Not Alter Synaptic Marker Levels in the Striatum and Cortex

Synuclein functions are closely linked to synaptic transmission, and its disruption could potentially underlie the observed phenotypes. Therefore, we analyzed several synaptic proteins involved in neurotransmission. We examined the dorsal striatum and cerebral cortex—key structures responsible for controlling motor and cognitive functions. Tissue samples from 13-month-old animals were analyzed by Western blotting.

Synaptophysin, a synaptic vesicle marker involved in exocytosis [48,49,50], is widely used as an indicator of synaptic density. Synaptophysin levels in both striatum and cortex did not differ significantly between KO and WT mice (Figure 9 and Appendix A).

Synapsin II is a neuron-specific phosphoprotein that maintains the reserve pool of neurotransmitter vesicles and mobilizes them when necessary, especially during high-frequency stimulation [51,52,53]. The IIa isoform of this protein is the principal regulator of vesicle dynamics and has a higher affinity for neurotransmitter vesicles. The quantity of synapsin IIa showed no genotype-related differences between KO and WT (Figure 9 and Appendix A).

Levels of Amphiphysin, which plays a major role in synaptic vesicle endocytosis [54,55,56], also did not differ among groups (Figure 9 and Appendix A). Taken together, these findings indicate that the absence of synucleins does not significantly affect synapse quantity, vesicle turnover, or synaptic plasticity in the examined brain regions suggesting preserved synaptic structure. In addition, we did not detect any differences in the levels of tyrosine hydroxylase (TH), a marker of dopaminergic neurons, in either the striatum or cortex (see Appendix A), which indicates no evident alterations in the dopaminergic system.

## 4. Discussion

The results obtained in this study reveal complex and multidirectional patterns of behavioral changes in γ-synuclein KO mice. Despite high homology and overlapping expression patterns among α-, β-, and γ-synucleins, our findings indicate that these proteins possess specialized, nonredundant functions that manifest under specific physiological conditions.

One of the most striking observations was lifespan reduction in γ-synuclein-deficient mice (γ-synuclein KO and αβγ-synuclein KO), whereas double αβ-synuclein KO mice displayed normal survival (Figure 1). Earlier work conducted on C57BL/6-background knockouts did not detect lifespan differences [28], but that study was limited to a 2-year observation window (~750 days). Extending the observation period to 3 years (>1000 days) in the present study revealed a late-onset phenotype, suggesting that γ-synuclein may uniquely influence long-term survival and that this effect becomes apparent only in advanced age.

A prominent characteristic of γ-synuclein deficiency was reduced muscle strength consistently observed in young (6-month-old) and middle-aged (13-month-old) mice (Figure 2 and Figure 3). However, this deficit was no longer distinguishable by 24 months from the age-related deficit in WT mice. The early reduction in muscle strength may reflect the importance of γ-synuclein in synaptic transmission in central and neuromuscular synapses [37]. The disappearance of differences in older animals could indicate physiological compensatory mechanisms developing with age, possibly mediated by other synuclein family members [24,57]. Supporting this, mice lacking all three synucleins displayed reduced grip strength across all ages, including 24 months, suggesting a failure of compensatory mechanisms in the complete absence of synucleins (Figure 2A, Figure 3A and Figure 4A). Further investigation is required to clarify the potential compensatory roles of other synucleins in the absence of γ-synuclein.

In the inverted grid test, evaluating endurance and motor coordination along with muscle strength, KO mice displayed a progressive, age-dependent decline (Figure 2B, Figure 3B and Figure 4B). This may suggest gradual accumulation of subtle deficits in motor regulation detectable with age progression. However, differences between groups diminished with age in other assays, such as grip strength. These opposing results suggest that disruption of synuclein function does not produce a straightforward cumulative phenotype but instead triggers complex, multidirectional compensatory processes within the nervous system. Our findings partially differ from results reported by Connor-Robson et al. [28] who found no decreased performance of 24-month-old γ-synuclein KO mice in the inverted grid test. But our findings are consistent with their observations for αβ-synuclein KO and αβγ-synuclein KO animals. Potential explanations for this discrepancy include subtle variations in diet, environmental conditions, or genetic drift within the C57BL/6 mouse line.

In contrast, γ-synuclein deficiency did not affect performance on the accelerating rotarod—a task that is more reliant on coordination and motor learning than muscle strength. In all age periods, γ-synuclein KOs did not differ from controls. A decrease in latency to fall was noted for the αβ-synuclein KO group at 6 months compared to WT, but these differences converged as the animals aged (Figure 2C, Figure 3C and Figure 4C). The triple KOs in our study performed significantly better than WTs at 24 months, in contrast with the decreased performance reported earlier [28]. We observed this enhanced performance in αβγ-synuclein KO mice across all age groups and attribute it to the general hyperactivity of these animals, previously reported and evident in other behavioral tests [27]. A key methodological distinction of the triple KO line used in this study is the use of a different α-synuclein KO allele [36] which lacks the unintended disruption of the adjacent *Mmrn1* gene occuring in the previously described line [58]. Direct comparisons using both α-KO lines are required to clarify these differences.

Exploratory and cognitive functions were assessed for middle-aged mice at 13 months. At this age, the motor phenotype was already well developed, but age-related deficits had not yet fully emerged. The most prominent behavioral alterations included reduced anxiety-like behavior in γ-synuclein KO mice and increased exploratory activity in αβ- and αβγ-synuclein KO mice in the open-field test (Figure 5 and Figure 8). γ-synuclein KOs consistently spent more time in the open arms of the elevated plus maze indicating diminished aversion to stress-associated stimuli. A similar reduction in anxiety-like behavior was observed in the synuclein null group. Nevertheless, this reduction was accompanied by a pronounced hyperactivity phenotype manifested by increased locomotor activity (Figure 8C). Across multiple behavioral assays, augmented locomotor activity was characteristic of αβ- and αβγ-synuclein KO mice, whereas reduced anxiety-like behavior was specifically associated with γ-synuclein deficiency. These findings suggest that γ-synuclein uniquely contributes to modulating anxiety-related behaviors, while locomotor hyperactivity emerges only when the other two synucleins or all three family members are absent.

Assessment of short-term and long-term memory in the Y-maze and Morris water maze tests revealed no significant cognitive impairments in any KO group (Figure 6 and Figure 7). Although KO mice exhibited slower learning during early water maze training, this likely reflects compromised motor function, reduced motivation, altered aversion to water or changes in the emotional state rather than memory dysfunction. This interpretation is consistent with prior studies showing that synuclein deletion influences stress reactivity, dopaminergic signaling, and initiation thresholds for behavioral responses [33,59,60,61]. Reduced anxiety in γ-synuclein KO mice previously documented in younger animals [33,62] appears to be a stable phenotypic manifestation of γ-synuclein loss.

A particularly notable finding is the preserved expression of synaptic markers (synaptophysin, synapsin IIa, and amphiphysin) in both the striatum and cerebral cortex across all KO groups (Figure 9). These results indicate that synaptic density and vesicular neurotransmitter turnover remain intact despite the absence of synucleins. Previously published data reported decreased dopamine and amphiphysin levels in aged α-synuclein KO mice but not in γ-synuclein KO mice [63]. In the present study, we confirmed that 13-month-old γ-synuclein KO mice show no changes in synaptic marker levels. Moreover, no differences were detected in both the striatum and cerebral cortex of same age αβ-synuclein KO or αβγ-synuclein KO mice. Thus, the behavioral and motor deviations observed in the KO lines are more likely to result not from structural synaptic abnormalities but from more subtle physiological alterations in neurotransmission dynamics.

The limitations of this study include the use of constitutive knockout mice, in which the corresponding synuclein genes are nonfunctional throughout the entire lifespan, including embryonic development. This may lead to formation of compensatory mechanisms due to high developmental plasticity. In this context, conditional knockout of synucleins in adult mice would represent a more relevant model, as it would better mimic the loss of protein function resulting from pathological protein aggregation during human neurodegenerative diseases. Moreover, although we did not detect changes in major synaptic protein markers, we cannot exclude the presence of more subtle synaptic abnormalities. Therefore, further comprehensive study of synaptic structure and function in synuclein knockout mice is needed.

## 5. Conclusions

In summary, our phenotypic analysis of γ-synuclein KO mice compared to mice that retain γ-synuclein but lack the other two family members (α- and β-synuclein) and to mice completely lacking all three synucleins indicates that γ-synuclein is crucial for longevity and motor system function. Loss of γ-synuclein leads to a sustained reduction in muscle strength. Another phenotypic consequence of γ-synuclein knockout in mice is change in anxiety-like behavior leading to a diminished response to aversive stimuli. This may suggest inability to adequately evaluate danger in the environment. At the same time, no other impairments of higher nervous functions, such as altered exploratory activity or memory deficits, were observed. In the absence of all three synucleins, some behavioral changes replicate changes in γ-synuclein KO mice, such as decreased muscle strength and reduced anxiety. However, triple KOs also exhibit additional abnormalities, such as hyperactivity, which is not present in other KO groups. Notably, even the complete absence of all three synucleins does not lead to overt physiological failure, yet it results in dysregulation of distinct, not directly linked behavioral aspects, such as motor performance and anxiety. These findings support the existence of specialized, non-overlapping functions of different synucleins and highlight a distinct role of γ-synuclein in regulating both motor performance and behavioral responses [33,64,65].

## Figures and Tables

**Figure 1 biomedicines-14-00092-f001:**
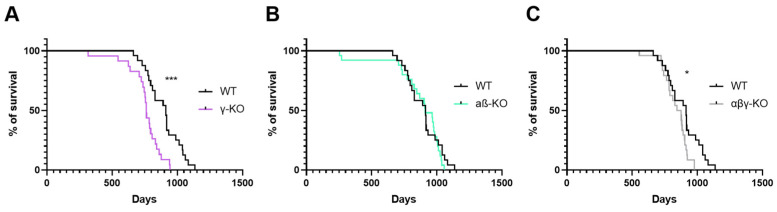
Survival curves of synuclein knockout and WT mice. (**A**) γ-Synuclein KO mice (γ-KO). (**B**) Double α- and β-synuclein KO mice (αβ-KO). (**C**) Triple synuclein KO mice (αβγ-KO). Statistical analysis: Mantel–Cox (log-rank) test. *—*p* < 0.05; ***—*p* < 0.001.

**Figure 2 biomedicines-14-00092-f002:**
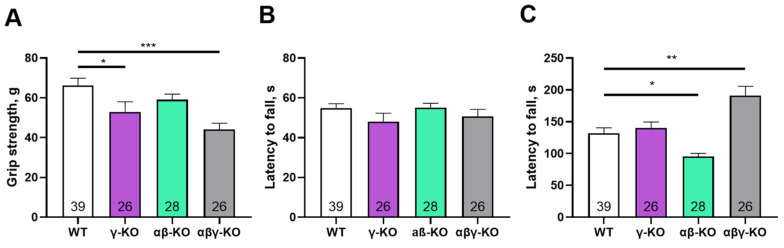
Motor performance of 6-month-old synuclein KOs (γ-KO, αβ-KO and αβγ-KO) and WT mice. (**A**) Grip strength test. (**B**) Inverted grid test. (**C**) Accelerating rotarod. Data are presented as mean ± SEM. Statistical analysis: Kruskal–Wallis test with Dunn’s multiple comparisons test. * *p* < 0.05; ** *p* < 0.01; *** *p* < 0.001. Sample sizes are indicated by the numbers within the bars.

**Figure 3 biomedicines-14-00092-f003:**
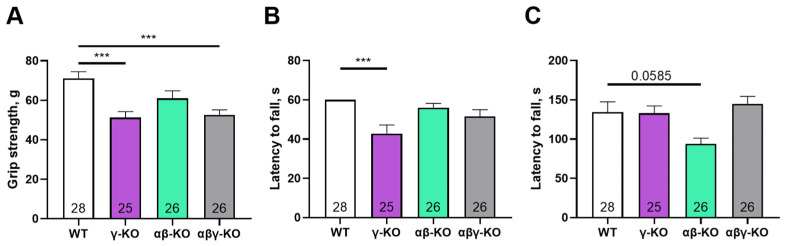
Motor performance of 13-month-old synuclein knockout (γ-synuclein KO, αβ-KO and αβγ-KO) and WT mice. (**A**) Grip strength. (**B**) Inverted grid. (**C**) Accelerating rotarod. Data are presented as mean ± SEM. Statistical analysis: Kruskal–Wallis test with Dunn’s multiple comparisons test. *** *p* < 0.001 or the exact *p* values are shown. Sample sizes are indicated by the numbers within the bars.

**Figure 4 biomedicines-14-00092-f004:**
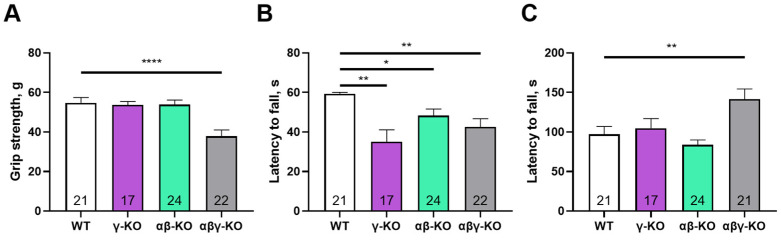
Motor performance of 24-month-old synuclein KO (γ-KO, αβ-KO and αβγ-KO) and WT mice. (**A**) Grip strength. (**B**) Inverted grid. (**C**) Accelerating rotarod. Data are presented as mean ± SEM. Statistical analysis: Kruskal–Wallis test with Dunn’s multiple comparisons test (**B**); one-way ANOVA with Dunnett’s correction (**A**,**C**). * *p* < 0.05; ** *p* < 0.01; **** *p* < 0.0001. Sample sizes are indicated by the numbers within the bars.

**Figure 5 biomedicines-14-00092-f005:**
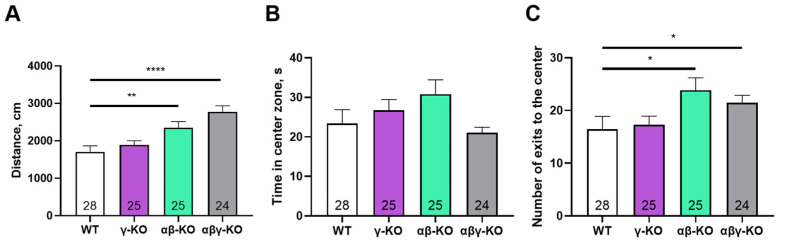
Open field test for 13-month-old synuclein KO (γ-KO, αβ-KO and αβγ-KO) and WT mice. (**A**) Average distance traveled. (**B**) Time spent in the center. (**C**) Number of center entries. Data are presented as mean ± SEM. Statistical analysis: Kruskal–Wallis test with Dunn’s multiple comparisons test (**B**,**C**) and one-way ANOVA with Dunnett’s test (**A**). * *p* < 0.05; ** *p* < 0.01; **** *p* < 0.0001. Sample sizes are indicated by the numbers within the bars.

**Figure 6 biomedicines-14-00092-f006:**
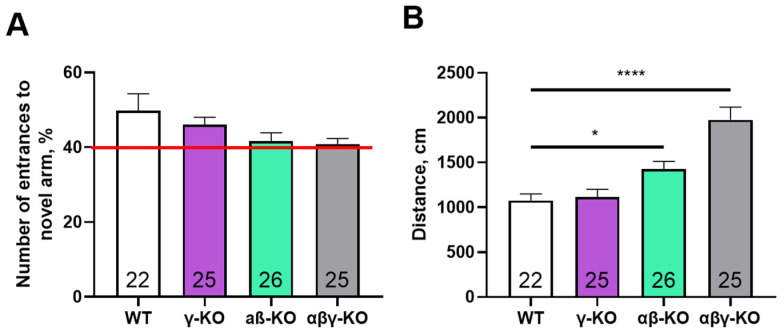
Y-maze test for 13-month-old synuclein knockout (γ-synuclein KO, αβ-synuclein KO and αβγ-synuclein KO) and WT mice. (**A**) Fraction of novel-arm entries. The red line indicates the 40% level, which is considered the minimal threshold for intact short-term memory. (**B**) Total distance traveled. Data are presented as mean ± SEM. Statistical analysis: Kruskal–Wallis test with Dunn’s multiple comparisons test (**A**) and one-way ANOVA with Dunnett’s test (**B**). Symbols: *—*p* < 0.05; ****—*p* < 0.0001. Sample sizes are indicated by the numbers within the bars.

**Figure 7 biomedicines-14-00092-f007:**
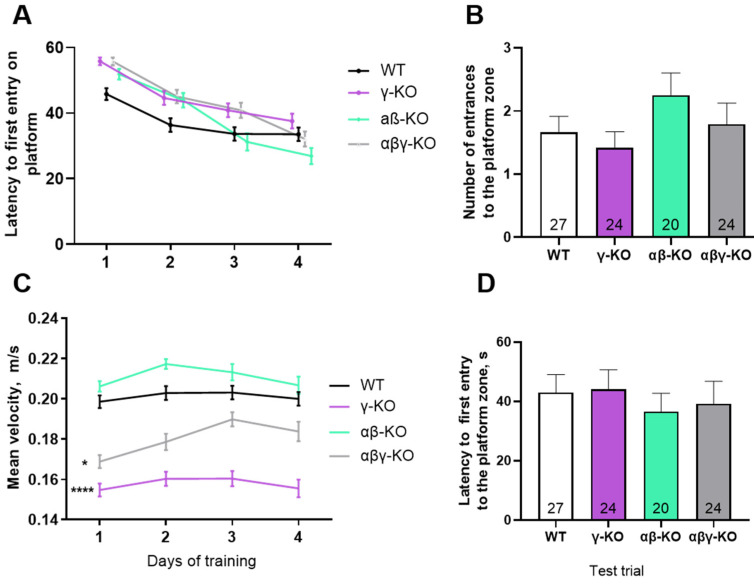
Morris water maze test for 13-month-old synuclein KO (γ-KO, αβ-KO and αβγ-KO) and WT mice. (**A**) Latency to find the platform across 4 training days. (**B**) Number of platform-zone crossings during the day-5 probe trial. (**C**) Average swimming speed across 4 training days. (**D**) Latency to first platform-zone entry at the day-5 probe trial. Data are presented as mean ± SEM. Statistical analysis: Kruskal–Wallis test with Dunn’s multiple comparisons test (**B**,**D**) and two-way ANOVA with Dunnett’s test (**A**,**C**). Symbols: *—*p* < 0.05, comparison between WT and γ-KO at all time points; ****—*p* < 0.0001, comparison between WT and αβγ-KO at all time points. Sample sizes are indicated by the numbers within the bars (**B**,**D**).

**Figure 8 biomedicines-14-00092-f008:**
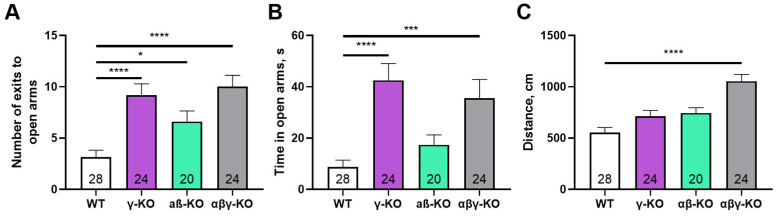
Elevated plus maze test for 13-month-old synuclein KO (γ-synuclein KO, αβ-synuclein KO and αβγ-synuclein KO) and WT mice. (**A**) Number of entries into open arms. (**B**) Time spent in open arms. (**C**) Total distance traveled. Data are presented as mean ± SEM. Statistical analysis: Kruskal–Wallis test with Dunn’s multiple comparisons test (**A**,**B**) and one-way ANOVA with Dunnett’s test (**C**). Symbols: *—*p* < 0.05; ***—*p* < 0.001; ****—*p* < 0.0001. Sample sizes are indicated by the numbers within the bars.

**Figure 9 biomedicines-14-00092-f009:**
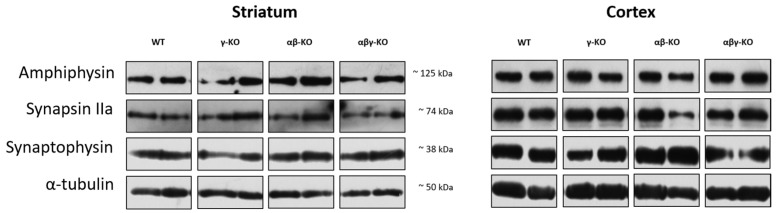
Synaptic proteins in the brain of 13-month-old synuclein KO (γ-KO, αβ-KO and αβγ-KO) and WT mice, representative Western blots.

## Data Availability

Data are contained within the current article and its Appendix A.

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
