# Peer review of "Distinct Role of γ-Synuclein in the Regulation of Motor Performance and Behavioral Responses in Mice"

_biomedicines, 2026, doi:10.3390/biomedicines14010092_

Round 1
Reviewer 1 Report
Comments and Suggestions for Authors
The manuscript by Michail S. Kukharsky et al., entitled “Distinct role of γ-synuclein in the regulation of motor performance and behavioral responses in mice” have described the distinct role of gamma synuclein in regulating motor performance and behavioral responses. The experimental methods are well described. Overall, the manuscript is well written, and addressing the minor suggestions listed below will further improve its clarity and quality.
1. I would recommend including a Western blot for tyrosine hydroxylase for all conditions to evaluate dopaminergic innervation of the striatum. This would provide information about the role of gamma-synuclein knockout in dopaminergic innervation.
2. I would recommend including immunohistology analysis of brain regions (striatum and cortex) against tyrosine hydroxylase antibody for all four conditions.
3. In Figure 2,3,4,5,6 please specify the meaning of the numbers shown inside each bar, as their significance is not currently clear.
4. In Figure 7B and 7D, please specify the meaning of the numbers shown inside each bar, as their significance is not currently clear.
5. For plots in which no statistically significant differences are observed, please include the notation ‘ns’.
6. I recommend performing Western blot analysis for muscle structural markers (e.g., dystrophin, desmin) under all experimental conditions, as this would help evaluate the contribution of gamma-synuclein to motor performance.
Reviewer 2 Report
Comments and Suggestions for Authors
This manuscript presents a study did comparative examination of synuclein knockout conditions between γ, αβ and triple KO conditions by emphasizing to interrogate γ- synuclein role. The introduction is relevant, and the methodology is appropriate. The critical analysis of the manuscript and related queries are listed below:
1) Authors should clarify representation or explanation of latency to fall (s) in figure 2, 3 and figure 4 because it seems confusing. As per the data, γ synuclein KO condition influences muscle strength and αβ KO condition influences motor coordination or learning. However, it is difficult to understand that the triple KO improves the motor coordination and learning despite impaired muscle strength (γ synuclein KO) and impaired motor coordination (αβ KO). Indeed, it may worse the motor coordination having much higher numbers of fall within shorter time (second) spent on rotarod as compared to the wildtype condition. If it improves motor coordination and learning, then what should be the mechanism helps triple KO to outperform the wildtype condition in absence of all three synuclein.
2) In figure 4, I wonder to understand that γ synuclein KO conditions do not have any aging effect on muscle strength as WT has but the KO condition reduced life span. If this KO does not have aging effect on muscle strength, then there should be any influence of γ synuclein on aging.
3) Authors missed to show influences of KO conditions on dopamine level. It is also better to understand that it helps to correlate with behavioral test results and previous findings in similar context.
4) In the discussion, Author stated two plausible things. One is “The early reduction in muscle strength may reflect the importance of γ-synuclein in synaptic transmission in central and neuromuscular synapses”. Authors may consider studying the synaptic protein levels at the early age to just confirm whether there is any influence of γ-synuclein KO on synaptic proteins at early age then it will improve the impact and strength of the present study.
Second statement is “The disappearance of differences in older animals could indicate physiological compensatory mechanisms developing with age, possibly mediated by other synuclein family members”. If αβ provides compensatory mechanisms with developing age, then what is the reason that αβ KO did not show any aging effect or deficits.
Reviewer 3 Report
Comments and Suggestions for Authors
The work is interesting and promising, but substantial clarifications, additional analyses, and improved reporting are required.
-The Methods state “each experimental group included at least 20 males” and “exact sample sizes are indicated in the figure legends”, but there are not clear Ns per test/age/analysis. Please report sample sizes in all figure legends.
-The study measures the same genotypes at 6, 13 and 24 months. For measures like grip strength and inverted grid, please analyze with a two-way ANOVA / mixed model (age × genotype), report interaction terms and post-hoc contrasts. This will clarify whether genotype effects are age-dependent (the text claims that but statistics shown do not formally test interaction).
-All experiments used males (Methods). Sex strongly modulates synuclein biology and behavior. Either (A) provide female data or (B) give a strong rationale and discuss this limitation explicitly.
-γ-KO and αβγ-KO mice swam slower; this confounds latency measures. Authors should re-analyze spatial memory with swim speed as a covariate (ANCOVA) or use distance to platform / platform crossings as primary endpoints and show that memory conclusions are robust after accounting for motor differences. The current interpretation risks conflating motor/anxiety and memory.
-The manuscript suggests γ-synuclein is “essential for neuromuscular function” based mainly on grip strength and hanging. That is a strong mechanistic claim without direct neuromuscular or peripheral data. I recommend either performing (or adding if available) one or more of: electromyography (EMG), neuromuscular junction morphology (e.g., α-bungarotoxin staining), muscle histology, or measurement of peripheral nerve conduction; or else temper language to state a functional association and propose mechanisms as speculative.
-The authors note the triple KO uses a different α-KO allele that avoids disruption of Mmrn1 (Discussion). This is important and may explain differences with prior studies. Provide genotyping/sequence confirmation and explicitly describe which α-KO allele was used in each line and any backcrossing performed. If possible, include control experiments or references that demonstrate the absence of off-target disruptions. This is critical for interpreting the phenotype as due to synuclein loss rather than background effects.
Minor points
-State how many independent animals per genotype were pooled or run separately.
Comments on the Quality of English Languagemoderate editing
Reviewer 4 Report
Comments and Suggestions for Authors
This manuscript presents a phenotypic analysis of γ-synuclein knockout mice compared with αβ-synuclein double knockout, αβγ-synuclein triple knockout, and wild-type controls. The study addresses an important and unresolved question regarding the non-redundant roles of individual synuclein family members in motor, behavioral, and survival phenotypes. Several conceptual, methodological, and interpretational issues should be addressed to strengthen the conclusions, particularly regarding mechanisms, statistical handling of behavioral confounds, and clarity of data presentation.
1- Why did the authors determine the markers in the striatum and cortex (and not the midbrain)?
2- Have all mice in each group been subjected to all seven behavioural tests mentioned?
3- The mice deaths were spontaneous or euthanasia-related?
4- Mention the limitations of the study at the end of the 'discussion' section.
5- Correct the typo errors in the manuscript.
6- The authors should re-organize the 'discussion' section. It should be summarized and should not be repetitive with the 'results' section.
Round 2
Reviewer 3 Report
Comments and Suggestions for Authors
the authors clarified several issues.